# Phytoremediation as an Effective Remedy for Removing Trace Elements from Ecosystems

**DOI:** 10.3390/plants12081653

**Published:** 2023-04-14

**Authors:** Agnieszka Mocek-Płóciniak, Justyna Mencel, Wiktor Zakrzewski, Szymon Roszkowski

**Affiliations:** 1Department of Soil Science and Microbiology, Poznan University of Life Sciences, Szydłowska 50, 60-656 Poznan, Poland; 2Regional Chemical and Agricultural Station in Poznan, Sieradzka 29, 60-163 Poznan, Poland; 3Department of Geriatrics, Ludwik Rydygier Collegium Medicum in Bydgoszcz, Nicolaus Copernicus University in Torun, Jagiellonska 13/15, 85-067 Bydgoszcz, Poland

**Keywords:** trace elements, contaminated soils, phytoremediation, phytoextraction, phytostabilization, hyperaccumulator, rhizofiltration, plant endophytes, microorganisms

## Abstract

The pollution of soil by trace elements is a global problem. Conventional methods of soil remediation are often inapplicable, so it is necessary to search intensively for innovative and environment-friendly techniques for cleaning up ecosystems, such as phytoremediation. Basic research methods, their strengths and weaknesses, and the effects of microorganisms on metallophytes and plant endophytes resistant to trace elements (TEs) were summarised and described in this manuscript. Prospectively, bio-combined phytoremediation with microorganisms appears to be an ideal, economically viable and environmentally sound solution. The novelty of the work is the description of the potential of “green roofs” to contribute to the capture and accumulation of many metal-bearing and suspended dust and other toxic compounds resulting from anthropopressure. Attention was drawn to the great potential of using phytoremediation on less contaminated soils located along traffic routes and urban parks and green spaces. It also focused on the supportive treatments for phytoremediation using genetic engineering, sorbents, phytohormones, microbiota, microalgae or nanoparticles and highlighted the important role of energy crops in phytoremediation. Perceptions of phytoremediation on different continents are also presented, and new international perspectives are presented. Further development of phytoremediation requires much more funding and increased interdisciplinary research in this direction.

## 1. Introduction

As a result of the ongoing industrialization of the world, which undoubtedly brings considerable economic benefits, the pollution of the natural environment has increased significantly. Soil is the largest reservoir of chemical pollutants, including trace elements, and it is a key element in the soil-plant-animal-human trophic chain [1]. Therefore, the pollution of soils with trace elements (TEs) and metalloids poses a threat to the normal function of the pedosphere. TEs are metallic elements with a density of more than 4.5 g·cm^−3^. They are characterized by relatively high atomic weights and atomic numbers [2]. They have adverse effects on living organisms and, when in excess, block basic life processes [3,4,5,6]. TEs do not decompose in biological and physical processes; therefore, they persist in the soil and present a long-term (thousands of years) environmental threat [7,8,9,10,11,12,13]. For example, lead (Pb) can persist in soil for more than 150–5000 years and remains at high concentrations for up to 150 years after sludge application to the soil [14,15], whereas the biological half-life of cadmium (Cd) is approximately 10–30 years [15,16]. Therefore, TEs are an important factor limiting the abundance, activity and biodiversity of microorganisms and plants [8,17]. Their sources can be divided into natural and anthropogenic [2,18]. TEs can come from two sources—natural (products of bedrock weathering, volcanic eruptions, ocean evaporation, forest fires) and anthropogenic (mining, metallurgy, municipal and household waste, sewage discharges, industrial and commercial activities, oil industry, warfare, nuclear power plants, use of agrochemicals, active and inactive military zones—weapons testing, bomb disposal, shooting exercises) [9,11,19,20,21,22,23,24,25,26,27,28,29,30,31,32,33].

TEs are persistent inorganic chemicals. They have cytotoxic, genotoxic and mutagenic effects on plants [34]. We can divide these elements into: essential micronutrients for plants (Cu, Fe, Mn, Mo, Ni and Zn), non-essential elements or toxic, even in small amounts, elements for plants (As, Cd, Co, Cr, Hg, Pb, Sb, Cr) [35,36,37,38,39,40,41,42,43,44,45,46,47,48,49,50,51]. TEs can limit important processes such as enzymatic activity and photosynthesis [52]. These negative impacts occur because metals disrupt regular metabolic pathways in plants [53]. Micronutrients are usually components of enzymes and other proteins crucial to metabolic processes. When the concentration exceeds the threshold value, these TEs become toxic for plants. For example, excess arsenic (As) causes photosynthesis inhibition and decreases biomass and yield. Cadmium (Cd) is a highly toxic TE due to its fast mobility and persistency. A very small concentration of Cd is lethal to plants [37]. Cadmium toxicity causes chlorosis, reduced water and nutrient uptake, browning of root tips, and ultimate death. Chromium (Cr) and lead (Pb) stress cause reduced nutrient uptake and disturbance in metabolic pathways, respectively. Mercury (Hg) and zinc (Zn) toxicity cause reduced photosynthesis due to the inhibition of photosystems I and II. Furthermore, excess nickel (Ni) causes retarded seed germination, reduced plant height, reduced root length, and also reduced chlorophyll content [37].

Plants, to defend themselves from the negative effects of TEs, use their defense systems [54]. At the very beginning, plants use an avoidance strategy, which involves limiting the uptake of TEs or blocking their access to the root. This can involve sequestration, immobilization or complexation of metals through root exudates [37,55]. If the previously mentioned defense systems are not sufficient, plants activate TEs tolerance mechanisms, such as metal ion trafficking, metal binding, metal chelation, accumulation of osmolytes and osmoprotectants or intracellular complexation [56]. However, the presence of significant amounts of TEs in the soil inhibits the development and activity of microorganisms, which leads to the disruption of processes related to the decomposition and transformation of organic matter [57,58]. The deficit of soil microbes and humic compounds contributes to ionic imbalance and increases the pool of bioavailable forms of TEs in the soil sorption complex [59]. There are various methods of cleaning up an environment contaminated with TEs. Conventional physicochemical methods require significant financial resources and most often involve the complete replacement of the contaminated soil layer [60,61]. These methods are also energy-intensive and produce large amounts of toxic waste [62,63]. The cost of conventional methods is estimated at $10–1000 per m^3^ of soil. Methods that involve the treatment of soils with plants are much cheaper (about $0.05 per m^3^ of soil) and more effective [64]. The degree of bioaccumulation (the accumulation of harmful substances in the plant) depends on various factors, e.g., the TEs content in the soil, the organic matter content, the soil type and structure, soil moisture, soil pH and the plant species [65]. There is a group of plants that have developed a number of mechanisms (e.g., polypeptides called phytochelatins) that allow TEs to accumulate in their tissues [64]. However, the synthesis of phytochelatins depends on the plant organ, the duration of exposure and the concentration of metal in the medium. It is also worth mentioning that this process is associated with slower plant growth [66,67,68,69]. Therefore, it is important to look for innovative solutions to clean up endangered ecosystems. Biological methods are becoming increasingly important. Numerous scientific studies have shown that certain plant species, thanks to their specific characteristics, have both the ability to take up and degrade xenobiotics polluting the environment.

The effectiveness of phytoremediation for the treatment of heavily contaminated soils is generally low, as the plants used take up a hundred percent small amounts of TEs. This would require their use for hundreds of years, with the reduction or complete removal of emission sources. On the other hand, they can be effective for the reclamation of less polluted soils located along traffic routes or parks, squares and urban greenery, i.e., places of frequent residence of various age groups. The idea of using plants to reduce and level pollution in the environment has been known for a long time. In addition to aesthetic value, protection from noise, providing oxygen, plant species with high phytoremediation abilities planted in urban areas (maple leaf plane, Japanese larch, poplar, ash, field maple, white and sessile dogwood, wrinkled rose, common yew), have the opportunity to play a health-promoting role. This is because they contribute to a significant improvement in the urban environment in which we live. The ability of plants to take up TEs and accumulate PAHs and particulate matter (products of traffic pollution) in the wax overhang makes phytoremediation a very attractive technology dedicated to urban areas. Plants with phytoremediation capabilities act as a “green liver” in the urban environment.

From the literature collected for this review article, it appears that research to improve and refine phytoremediation methods is practiced and actualized, but further steps in this direction are still needed. In practice, the use of only one method or treatment for effective phytoremediation will not be sufficient or satisfactory. Plant-microbiome interactions are proving to be an extremely effective approach for TE uptake and translocation in plants. Our work holds high hopes for further exploration of new metabolites and pathways for the efficient degradation of contaminants through the plant-microbiota system. With modern bioengineering techniques, it is possible to modify plants with desirable traits, as well as to isolate microorganisms and then introduce them into the soil to improve phytoremediation using appropriate plant species. In-depth and interdisciplinary research in this direction with significantly increased funding is needed in order to obtain, through these modifications, both plants and microorganisms that are effective in the remediation of contaminated land and, in addition, resistant to difficult and often changing environmental conditions.

For this literature review, papers from 2000–2023 were used. Older papers were used only for the clarification of terms. Data were searched in Scopus, PubMed, Web of Science, ScienceDirect, Public Library of Science and AGRO databases. Search engines such as Google Scholar, MDPI Search and ResearchGate were also used. Searches were mainly conducted by using key-words, synonyms, combining terms and database search limits, e.g., source type and topic.

## 2. The Essence of the Process of Phytoremediation

The term *phytoremediation* is a combination of the Greek word *phyton* (plant) and the Latin word *remediare* (repair) [70]. Phytoremediation involves the use of plants that can grow in a contaminated environment and influence the biological, chemical, and physical processes taking place in it to ultimately contribute to the effective removal of xenobiotics from the biological system [71,72,73,74,75]. About 400 ecotypes of metal-accumulating plants are known. They are called hyperaccumulators. Hyperaccumulators are plants that are characterized by the accumulation of particular metals in their tissues. What distinguishes them from other plants is that the concentration of metals accumulated in their tissues can be hundreds or thousands of times greater [2,76,77]. In order to call a plant a hyperaccumulator, if it is growing in its natural habitat, the concentration of metal in the shoot in the dry weight of the leaf tissue should be more than: 100 µg g^−1^ of Cd, Tl, Se; 300 µg g^−1^ of Co, Cu, Cr; 1000 µg g^−1^ of Ni, As, Pb or rare earth elements (REEs); 3000 µg g^−1^ of Zn; 10,000 µg g^−1^ of Mn [2,76,77,78].

Some plant species are more suitable for phytoremediation than others. There are two coefficients worth mentioning, including the bioconcentration factor (BCF) and translocation factor (TF). The BCF determines the ratio of metal content in the plant to that in the soil. If the BCF ≤ 1.00, it indicates the plant can only absorb but not accumulate metal. If the BCF value is higher than one, it means that the plant is a hyperaccumulator. TF determines plant efficiency in TEs translocation from the root to the shoot. TF higher than 1 indicates that the plant translocated TEs effectively from root to shoot. TF < 1, however, indicates ineffective metal transfer suggesting that these types of plants accumulate metals in the roots and rhizomes more than in shoots or the leaves. In hyperaccumulators, both ratios should be greater than unity [79,80,81,82,83,84,85,86,87]. Plants with high biomass and high bioaccumulation in aboveground parts of plants (BF_aboveground parts of plants_ > 1) are suitable for phytoextraction [83,88,89], while plants with a high bioaccumulation factor for belowground parts of plants (BF_belowground parts of plants_ > 1) and at the same time a low translocation factor (TF < 1) are suitable for phytostabilization [89,90]. Examples of hyperaccumulators and recommended phytoremediation methods are shown in Table 1.

A ratio also used to assess the distribution of TEs within the shoots and roots of plants is the root/shoot (R/S) ratio. This ratio indicates the concentration of TEs that is accumulated in the root to the concentration in the plant shoot. Plant roots are the final site of absorbed TEs, and shoots are able to accumulate smaller amounts than roots. Plants that are not hyperaccumulators have a shoot-to-root ratio of less than one. Hyperaccumulators should have a shoot-to-root ratio greater than one. This indicates efficient transport of TEs from roots to shoots. [87,143,144]. Hyperaccumulators and the TEs content of their parts are shown in Table 2.

Plants used in phytoremediation processes should tolerate high concentrations of xenobiotics, accumulate or biodegrade large amounts of contaminants, accumulate several contaminants simultaneously, grow rapidly, adapt easily, produce large amounts of biomass and be resistant to diseases, pests and harsh environmental conditions. In addition, it is worth noting the distinction between scientific and commercial purposes because the commercial success of the use of plants in contaminated environments depends on it. Plants are compared to solar-powered pumps, which extract specific elements from the environment and accumulate them in tissues [155]. Phytoremediation does not require the use of specialized and complex equipment. It is considered an effective, non-invasive, economical, socially acceptable, feasible method and an ecological alternative to physical remediation methods which interfere with the ecosystem [7,71,156]. For economic reasons, phytoremediation is especially used in many areas. One of them is degraded brownfields, where the primary goal of phytoremediation is to restore them to a state that is safe for development so that they can be used as recreational (sports), commercial or residential areas. The phytoremediation of brownfields consists of the cultivation of annual, pollution-tolerant plants, which give large amounts of biomass and are characterized by a high degree of environmental clean-up efficiency [157]. Phytoremediation is also applied in the vicinity of transportation routes, where it should be a continuous process so that pollutants can be removed on an ongoing basis. This area of phytoremediation is based on pollution-tolerant perennial plants, such as trees which have a large surface area and accumulate pollutants in the air [158].

Plants with a high phytoremediation capacity have also become increasingly popular in recent years and are being used in large urban areas to create green roofs (Table 3). Their purpose is to reduce the concentration of TEs in the ground but also to improve the quality of rainwater runoff [159].

However, research on the phytoremediation potential of green roofs is very limited [160]. Vijayaraghavan and Joshi [161] compared roof structures with and without vegetation; they found that green roofs provided both high-quality rainwater runoff and low concentrations of TEs. Moreover, Beecham and Razzaghmanesh [162] examined the quality of runoff from green roofs (*n* = 12) and without vegetation (*n* = 4) over a 12-month period. They found that pollutant concentrations were higher in runoff from systems without vegetation compared to green roofs. Thus, exemplary phytoremediators used on roofs have a great future, and research in this direction should be continued.

Currently, the term *phytoremediation* refers to several techniques employing higher plants to clean up contaminated ecosystems. These include rhizofiltration, phytoextraction, phytoevaporation and phytostabilization.

### 2.1. Rhizofiltration

The rhizofiltration process relies on the ability of the roots of selected plant species to absorb and adsorb pollutants from ground and surface waters, industrial, municipal, and agricultural wastewaters, as well as acid mine water [163]. The process of preparing plants for rhizofiltration involves growing them in clean water to develop large root systems; then, the plants are transferred from clean water to contaminated water to acclimate. After successful acclimatization, the plants are moved to the target contaminated site so they can remove TEs from there [164]. Rhizofiltration can be supported by symbiotic fungi and bacteria. The method is used to remove TEs ions as well as some organic substances and radioactive elements found at relatively low concentrations in aquatic environments. The resulting complexes are readily absorbed by plants. For example, the *Alyssum lesbiacum* plant uses histidine to complex nickel [165]. By acidifying the rhizosphere, plant roots cause TEs to become more available and take up these pollutants more efficiently. Pb is most effectively removed through rhizofiltration [166]. This process does not require an active biological system and also occurs on dead root tissue [167]. Various technical variants of rhizofiltration have been implemented, e.g., variants involving the use of mats floating on the surface of the water and keeping the roots of plants in water (*Helianthus* sp.), with aquatic plants such as *Phragmites australis* (Cav.) Trin. ex Steud., *Typha laifolia* (L.), *Eichhornia crassipes* (Mart.) Raf., *Lemna minor* (L.) [168,169]. Although the aquatic environment is the natural habitat of plants used in the rhizofiltration process, terrestrial plants are also gaining interest. Plants grown in hydroponic or aeroponic cultures remove contaminants more efficiently than aquatic plants [170]. Plants used in this method should not only have a dense root system and produce large amounts of biomass, but they should also exhibit high tolerance to TEs. For wetland remediation, it is common to use species characteristic of aquatic habitats, such as hyacinth, azolla, duckweed, cattail and poplar. These species meet the aforementioned requirements, such as high tolerance to TEs and high biomass [171]. Terrestrial plants are often characterized by a longer and hairier root system than aquatic plants. The following species are used for rhizofiltration: Indian mustard (*B. juncea*) and sunflower (*H. annuus*) [172,173].

### 2.2. Phytoextraction

Phytoextraction (or phytoaccumulation) is the use of plants to remove pollutants from water and soil and then place and accumulate them in their aboveground biomass [174]. The potential of hyperaccumulator plants is used to absorb sizable amounts of TEs. The technology consists in mobilizing ions by reduction with chelating compounds, the uptake of contaminants from the soil by plant roots, followed by transport in the xylem, redistribution to tissues and sequestration in cells [101,175,176,177]. Next, the vegetation is harvested and removed. The process can be repeated many times until satisfactory results are achieved, i.e., metals such as Cu, Cd, Cr, Pb, Ni and V are permanently removed [14]. The efficiency of this process depends on the choice of plants and the amount of water (along with the substances dissolved in it, e.g., heavy metals) passing through them per unit of time. It is noteworthy that this method has been proved to successfully remove TEs from the soil with plants such as *Helianthus annuus, Cannabis sativa, Nicotiana tabacum,* and *Zea mays* [178,179,180]. Grasses can also be used for phytoextraction, as they are characterized by a short life cycle, rapid biomass growth, and high tolerance to environmental stresses [181]. *Trifolium alexandrinum* is also a suitable plant for catching Cd, Pb, Cu and Zn. Like grasses, this plant species can be harvested a few times in one season because it grows quickly [59]. *Sebertia acuminata*—an endemic tree growing in New Caledonia—has a high potential for the phytoaccumulation of metals. It is a hyperaccumulator of nickel. Its latex-type sap contains about 25% Ni (about 11% by weight).

Phytoextraction-assisted chemicals are being used to increase the uptake of TEs by plants. This method includes the use of TEs chelators such as ethylenediaminetetraacetic acid (EDTA), N-hydroxyethyl-EDTA (HEDTA) or citric acid. These compounds increase the ability of plants to take up TEs and translocate them within plants (Table 4) [182,183].

Natural low-molecular-weight organic acids (citric acid, oxalic acid or vanillic acid) have been studied as alternative chelators to EDTA because of their rapid biodegradation rates [184,196]. Unfortunately, these chemicals can biodegrade rapidly, often leading to degradation before the metals are absorbed by plants [184]. However, further research in this area is required to find an alternative that offers the same results as EDTA. Although chelates have been used to aid phytoextraction and increase the recovery of TEs, such activities can have negative environmental impacts. There are still opportunities to develop green chemical technologies that increase the availability of elements to plants without damaging the environment [183].

### 2.3. Phytoevaporation

Phytoevaporation, also known as phytoxidation or phytovolatilization, involves the uptake of contaminants by plants, their transpiration and subsequent evaporation in a modified form. The process is primarily used to clean up aquatic environments and soils contaminated with selenium, mercury or arsenic [197]. Some organic compounds, such as trichloroethylene, benzene, nitrobenzene, phenol or atrazine, can also undergo phytoevaporation [198]. The best-known example of phytoevaporation is the remediation of selenium-contaminated environments. This element is most often found in the form of selenate (SeO_4_^2−^), selenite (SeO_3_^2−^) and occasionally in the organic form of selenomethionine. The rate of selenium uptake from the substrate depends on its chemical form and other factors, such as the concentration of SO_4_^2−^, which is a competitive ion, as well as the levels of glutathione and O-acetylserine in plant cells. When selenium is taken up, thanks to enzymatic reactions involving ATP sulfurylase, APS reductase, glutathione reductase, sulfite reductase, and S-methylmethionine hydrolase in chloroplasts, it is reduced to dimethylselenide (DMSe) or dimethyldiselenide (DMDSe) and released into the atmosphere. Both of these methylated forms of selenium (DMSe and DMDSe) are 500–700 times less toxic than the inorganic form of selenium [199,200]. As it is not easy to remove mercury from the aquatic environment, phytoevaporation is a promising technique for the remediation of this element. There are plants that can take up and accumulate mercury. However, they do not have the appropriate enzymes to catalyze the reduction of Hg^2+^ to Hg^0^. Therefore, genetic engineering techniques are hoped to solve the problem. Transgenic plants, such as radish *(Arabidopsis thaliana)* and tobacco *(Nicotiana tabacum),* contain bacterial genes for the enzymes—organic mercury compound lyase (MerB) and mercury reductase (MerA). They take up mercury (mainly in the methylated form) and reduce it to the elementary form Hg^0^ [71,198]. Phytoxidation is considered a rather risky method because the pollutants removed during this process enter the atmosphere. Although their form is less toxic, they still pose a serious threat to the ecosystem.

### 2.4. Phytostabilization

Phytostabilization is a process that does not involve the removal of TEs. Instead, they are retained in the soil through absorption and accumulation in the roots, adsorption on the root surface or precipitation in the rhizosphere, and thus, there is a lower environmental risk [157,201]. This process can occur through the absorption of TEs and their sequestration in the root tissues, adsorption on the root cell walls, and precipitation or reduction of metal valence in the rhizosphere [202,203,204]. The soil becomes physically stabilized, which counteracts water and wind erosion, whereas the vegetation cover is restored, which reduces the spread of metals into water or air [179,205]. The immobilization of TEs can be further assisted by the addition of organic matter in the form of biomass, sludge or composts, raising the pH value by liming the addition of carbonates or phosphates [14,206,207,208]. Phytostabilization is recommended for fine-grained soils with high organic matter content. Phytostabilization has an advantage over phytoextraction because the removal of hazardous biomass is not required [164]. Plants used for phytostabilization should have a high bioconcentration rate and a low rate of translocation of metals to the shoots. In addition, they should exhibit high tolerance to soil contamination and produce a sizable root biomass [101,201,208,209]. Phytostabilization can be supported by soil microorganisms such as bacteria and mycorrhiza. Thanks to them, the roots can increase their surface area and penetrate deeper into the soil. This facilitates phytostabilization and acts as a kind of barrier protecting the plant from the translocation of TEs ions from the roots to the shoots [210]. In addition, these microorganisms make heavy metal immobilization more efficient by adsorbing TEs on their cell walls, producing chelators, and promoting precipitation processes [211,212].

## 3. Benefits and Limitations of Phytoremediation

Like any method, apart from the numerous advantages, there are also some limitations to its use. The most important benefits of phytoremediation are [20,174,213,214,215,216,217,218,219]:the reduction of organic and inorganic pollution,the reduction of the amount of landfilled waste,the preservation and even improvement of the soil structure (compounds secreted into the rhizosphere by plant roots increase the population of microbiota in the soil, the pool of humic substance and soil fertility),the reduction of wind erosion by vegetation,no need for expensive, specialized equipment and personnel,the possibility of in situ application, which does not disturb the soil environment and prevents the spread of contaminants,lower cost than conventional remediation methods,the ease of implementation and maintenance (plants are a cheap, readily available and renewable source of energy),environmental friendliness and social acceptability,a lower noise level than that generated by other remediation methods (tree lagging reduces noise from industrial activities).

However, the use of phytoremediation is significantly limited by [20,174,213,214,220,221]:the depth of root penetration, the solubility and availability of contaminants,the longevity of the process—up to several decades,the scope of its application limited to areas with low and medium levels of pollution,special treatment of the biomass obtained by phytoextraction as a hazardous material,dependence on the climate and seasonality (the effectiveness of the process may be reduced due to damage to plants during the growing season, diseases, pests, and extreme weather conditions),avoiding the introduction of invasive and unsuitable plant species (foreign species disrupt biodiversity),the risk of transfer of metals to other environmental matrices such as water or air and inclusion in the food chain,the introduction of cultivation methods which can affect the mobility of TEs.

### Perceptions of Phytoremediation on Different Continents and New International Perspectives

Current research in phytotechnology is most often centered around genetics, physiology or biochemistry to increase plant tolerance and metabolism of both organic pollutants and TEs. In addition, efforts are being made to intensify the processes in the rhizosphere, which undoubtedly increase the phyto-availability of pollutants.

There are significant differences in the commercial management of phytoremediation on different continents [222]. In North America, private companies play a much larger role in phytoremediation investments. In their view, the process is seen as a “green revolution” in innovative technologies. In Europe, by contrast, the dominant approach is focused on solving phytoremediation problems and describing biological mechanisms. This could be overcome by spreading the aforementioned technology and gaining much greater public acceptance. In addition, standard remediation technologies, which are credited with more effective and longer-lasting performance, continue to enjoy strong support in Europe. Limited investment and ownership issues also contribute to this. In contrast, on the African or Asian continent, phytotechnologies are used on a larger scale than in many European countries regarding their commercialization and application. Due to the fact that they belong to technologies that do not bring far-reaching profits, they are classified as niche technologies. The further future of phytoremediation development must therefore involve the development of technologies for the utilitarian use of the biomass obtained. These problems at the current stage are very often overlooked or marginalized, except for the partial use of the obtained biomass for energy purposes.

The reasons for the intense search for effective hyperaccumulators around the world are many. One of them is the fact that they are, unfortunately, very locally found. In Poland, they are practically non-existent, except for one species that appears sporadically in Upper Silesia—*Arabidopsis halleri*. In other European countries, e.g., Germany, the Netherlands, or the Czech Republic, hyperaccumulators are usually very small plants. Among the most studied hyperaccumulators found in Europe is *Noccaea caerulescens* (alpine bollworts), the size of violets that appear on lawns in spring. In New Zealand, New Caledonia, and the Philippines, shrubs and trees are found, which in turn grow very slowly. The big obstacle is that either a plant takes up a lot of compounds but grows slowly (like hyperaccumulators), or it grows fast but takes up less (like energy plants). Therefore, intensive research is underway to improve the optimization of growing plants that will grow very fast and produce large biomass, and in the process—while taking up minerals from the soil—will also take up nuisance pollutants such as TEs from the soil. Depending on how much pollution has been accumulated in the plant, such plant biomass can be used differently. If the level of contaminants is very high, then the biomass can be harvested and burned. The resulting ash should then be stored as toxic waste, or it can be used for metal recovery or for the production of catalysts used in the chemical industry. Another way, with not-so-high levels of impurities in the plant, is to use biomass as fuel for heat or electricity generation or to produce biofuel. Then the impurities need to be separated in a technological process to ultimately produce clean biofuel [223].

Recently in Poland, great hope has been placed in the international project “GOLD” for optimizing the growth of three selected plant species (switch millet, industrial hemp and miscanthus) to achieve the greatest biomass and take up as much pollution as possible [224]. This innovative, international project, called “Bridging the gap between phytoremediation solutions on growing energy crops on contaminated lands and clean biofuel production,” has received sizable funding from Horizon in 2020. Maria Curie-Sklodowska University in Lublin (Poland) is a partner in the said project, coordinated by CRES—Centre For Renewable Energy Sources and Saving Fondation from Greece. This 4-year project is being carried out in a large consortium of 20 entities from the following countries: Greece, the Netherlands, Germany, China, Italy, France, Portugal, the United Kingdom, India and Canada. In the first stage of the project, the contaminated biomass will be pyrolyzed and gasified, resulting in vitrified ash containing toxic metallic impurities and syngas. In the second stage, on the other hand, the gaseous product will be converted into clean liquid biofuel. The collected biomass will be sent to the Netherlands and Germany, where specialized companies will test whether pure biofuel can be obtained from biomass produced in Poland, Greece, France, Italy, China and India. In addition, research will also be conducted to isolate contaminants from them that should not be found in such fuel. This project is very important because naturally polluted areas are being studied. This is because the above practices are translated into reality in two municipalities in Upper Silesia. Based on this research, universal strategies will be developed that can be applied to other potentially contaminated sites and used in various countries in the European Union and Asia. Such a diversity of analyzed research points (diversity in terms of climatic conditions or types of pollution) will allow the development of concepts that will be environmentally friendly as well as economically and socially rational for the production of clean biofuel. Thus, phytoremediation is becoming one of the elements of both an integrated and sustainable approach to the revitalization of polluted areas and the protection and shaping of the space in which we live.

Biofuel plants in phytoremediation have also been reported by Amin et al. [225]. According to the authors, of the plants tested (*Abelmoschus esculentus, Avena sativa, Guizotia abyssinica*, and *Glycine max*), *A. sativa* shows high Zn uptake, high tolerance and high biomass. This indicates that it is a suitable biofuel plant for both phytoremediation and biofuel.

Therefore, plant biomass converted into a renewable energy source represents an opportunity for phytoremediation plants on a global scale. It is worth noting that energy produced from plants accounts for 14% of global energy demand. Energy plants used for phytoremediation should be fast-growing, have large biomass and deep roots and yield an economically valuable product [226,227,228]. Table 5 shows energy plants used in phytoremediation serving as biofuel.

Studies on possibilities of the rational cultivation of chemically contaminated soils in the industrial sanitary Protection zones were conducted in Poland (Poznan University of Life Sciences). Humic deluvial soils (Regosols—IUSS-WRB, 2015), brown soils (Cambisols—IUSS-WRB, 2015) and proper black-earths (Phoaeozems—IUSS-WRB, 2015) occurring in the local depressions of the eastern part of the Copper Smelting Plant in Legnica (Lower Silesia) have been studied [230]. The soils were formed from the relatively small thickness of deluvial silt sediments. Heavy texture (granulometric composition) and relatively large amounts of organic matter have a decisive effect on the high geochemical resistance of the soils to Cu and Pb contamination. In addition to the elementary physical and chemical properties, total and available amounts of Cu, Pb and Zn, as well as total sulfur, have been determined. Moreover, different fractions of soil-copper have been isolated according to McLaren and Crawford’s method. The coefficients of correlation between many properties as well as the linear regression equations for some forms of soil-copper, have been calculated, taking into consideration only the amount of copper soluble in 1 mol/dm^3^ HCl. The plants showing resistance to high contents of Cu, Pb and S in soils and in biomass and thus suitable for the rational cultivation of the soils were found to be numerous varieties of shrub willow. The highest contents of Cu and Pb were determined in the leaves. The lowest contents of Cu and Pb were found in the stems, i.e., parts suitable for practical use (e.g., basketry purposes). In general, the American variety of willow in recommended for cultivation on these soils, whereas Piaskówka and Kottenheider are recommended for the more elevated areas.

The use of phytoremediation is key to achieving sustainable development. Plant-based methods provide a low-cost method of land remediation and are the best strategy for future use [10,231]. The increase in environmental pollution is prompting leaders and global institutions to take new steps to reduce the negative impact of TEs and their risks. New strategies must address current challenges and seek new, efficient solutions [232,233]. Nature-based solutions, or nature-inspired actions, are a solution to mitigate the environmental change in combination with economic, social and environmental benefits [234,235]. Research is still needed to develop new, effective methods for recovering metals from plant biomass.

## 4. Supporting the Processes of Bioremediation of Contaminated Soils

The threat posed by the accumulation of inorganic pollutants (TEs) in the environment is associated with the need to seek innovative, safe and unconventional methods to combat these pollutants [236]. Recent scientific developments suggest that bioremediation provides effective removal of xenobiotics using microorganisms, plants and enzymes. The continuous accumulation of pollutants in the environment means that microbes are not fully effective in protecting the environment. Hence, scientists around the world are turning their attention to the possibility of modernizing bioremediation methods by introducing microbial, organic and enzymatic preparations and substances to increase the effectiveness of biological remediation [236].

One of the methods supporting the remediation of contaminated sites is the use of sorbents in the first stage of soil remediation, additionally enriched with biomass. The task of sorbents is to inhibit the migration of hardly decomposable substances. Adding beneficial microorganisms to the sorbent supports bioremediation and can be a source of nutrients for them (a source of carbon), thus increasing the efficiency of the whole process. Bioremediation technology uses powdered materials with the properties of lignocellulose-based biosorbents (e.g., algae, the fungus *Trichoderma harzianum*), which have the property of adsorbing hexavalent chromium (which is toxic and water-soluble) and converting it to the trivalent form (insoluble in water) [237].

Wydro et al. [238] introduced an organic substrate in the form of municipal sewage sludge with low metal content into the soil. The study showed that plants in the sewage sludge-fertilized sites took up more Cd and Zn from the soil, as opposed to the control. In addition, biogenes—N and P—can be introduced to support the development of the microbiota. A prerequisite is that the sorbents used do not have a negative impact on the environment and that they are easily microbiologically degradable.

In bioremediation technology, it is also possible to apply bacterial strains that have the ability to produce surfactants—surface-active compounds. These compounds stimulate enzymatic processes, improving the bioavailability of contaminants, such as potentially toxic elements. Such surfactant-producing microorganisms include: *Bacillus megaterium* or *Pseudomonas aeruginosa* UG2. Surfactant-containing agents have been found to be used for rinsing contaminated grounds. The best results were obtained for flushing solutions containing cyclodextrins and rhamnolipids [239].

The increase in the number of compounds contaminating the environment has prompted the search for bioremediation methods using not only potential metabolites of microorganisms but also the enzymes themselves in the form of preparations. Such preparations may contain individual biocatalysts or enzyme complexes capable of changing toxic compounds into non-toxic ones. The use of enzymes as an aid to phytoremediation is believed to be advantageous because these compounds have a simple structure; moreover, the transformation of polluting compounds with the participation of enzymes does not result in the accumulation of toxic by-products, and the enzymes are utilized after the process by microorganisms residing in the polluted environment. Examples of bacterial enzymes that can take part in the remediation process are reductases, dehalogenases, monooxygenases or mono- and dioxygenases [240].

Nanoparticles may prove to be an innovative solution to aid the bioremediation process. These are particles with a size of 1–100 nm. Due to the fact that nanoparticles are a benign product for the environment, we can describe them as a method that carries potential environmental benefits. Macé et al. [241] conducted a study using a hydroxyapatite nanoparticle. The study showed that these particles reduced the availability of Cu and Zn in the soil. In turn, Khan and Bano [242] indicate that the use of nanoparticles improved the phytoremediation capacity of plants in relation to Cu, Zn, Ni and Pb. Adejumo et al. [243] applied silver nanoparticles (AgNPs) to *Zea mays* in their study. The results show improved shoot growth based on the root vigor index. AgNPs also increased the content of chlorophyll a and b and carotenoids; in addition, antioxidant activity increased. The authors point to improved phytostabilization of TEs while improving plant health values. However, there is a concern that nanoparticles used as a bioremediation aid may have a negative impact on the environment after a certain period of time, due to the possibility of releasing hazardous compounds. Some particles may have a bactericidal effect. Nanoparticles can be readily absorbed through cell membranes, with degradation having cytotoxic effects [244].

Brassinosteroids (BR) come in response to the negative impact of TEs on plant cells and the oxidative stress they cause. These are plant hormones that exhibit physiological activity in concentrations up to one hundred times lower than, for example, auxins. Due to their high biological activity, BRs regulate many processes in the plant. They can also reduce the toxicity of TEs. These hormones have the ability to regulate the absorption of trace element ions into cells and reduce the uptake of the above-mentioned elements through the roots, thanks to the high activity of the V-ATPase enzyme. Brassinosteroids also increase the activity of some antioxidant enzymes, which allows the removal of excess reactive oxygen species. In addition, BRs can stimulate the synthesis of phytochelatins that bind metal ions into complexes. Brassinosteroids play an important role in inducing plant defense mechanisms because they interact with other hormones, such as: auxins, cytokinins and salicylic acid [245,246,247].

Another method that supports phytoremediation is the use of transgenic plants. Plants that are used for this process should be characterized by a developed root system, rapid growth, production of large biomass and the ability to accumulate and tolerate very high concentrations of TEs. Therefore, genetic engineering can be used to create the ideal phytoremediation plant. An example is *Nicotiana glauca*; it was modified by a wheat gene encoding phytochelatin synthase (TaPCS1), resulting in potentially higher tolerance to Pb and Cd compared to a non-transgenic plant [248]. In contrast, transgenic *Brassica juncea* L. accumulated 1.5–2 times the concentration of Cd and Zn than wild Indian mustard [249]. Among trees, poplar is one of the excellent candidates for genetic engineering for phytoremediation. Poplar, introduced with the yeast cadmium factor 1 (ScYCF1) gene, has very high phytoremediation capabilities compared to non-transgenic poplar [250]. *Nicotiana glauca* with overexpression of the phytochelatin gene obtained from the *Thlaspi caerulescens* hyperaccumulator accumulated 24 times more Cd and 36 times more Pb [251]. The current state of knowledge suggests that the use of genetically modified plants makes it possible to clean soils contaminated with TEs. In addition to obtaining the above-mentioned plants, legislation and a general reluctance to use transgenic organisms may be a problem.

Phytoremediation allows the removal of metals from the soil and their accumulation in the above-ground parts of plants (phytoextraction) or immobilization in the soil at the root of the plant (phytostabilization). Some species of energy plants, growing on soils of lower quality and contaminated with TEs, successfully provide a yield sufficient for use on an industrial scale and enable their use in both processes [252]. Examples of energy plants that can be used in the phytoremediation process are: *Salix viminalis, Miscanthus × giganteus, Sida hermaphropdita.* Willow wood grown on soils heavily polluted with emissions from non-ferrous metal smelters may contain up to 4000 mg/kg Zn, 64 mg/kg Cd, 20 mg/kg Cu and up to 10 mg/kg Pb [253,254]. Kabala et al. [123] indicate that Miscanthus straw grown on unpolluted soils contains higher amounts of macronutrients, but lower amounts of TEs, than willow wood. In the case of mallow, concentrations are comparable to those in miscanthus straw. However, some studies indicate that mallow can more effectively clean the soil of Pb, Zn and Cu than willow. This is because it is more tolerant of soil contamination and has less yield reduction [123,255]. The basic aspects of growing energy crops are: the production of biomass as a source of renewable energy, utilization of sewage sludge for fertilization purposes, phytoremediation of chemically degraded soils. Therefore, it is necessary to obtain plants with high yields but also with higher phytoextraction abilities of TEs. This is where the previously described genetic engineering comes in, which, together with energy plants, can significantly improve and streamline the phytoremediation process [123].

Another method to support phytoremediation in TEs-contaminated waters is the use of microalgae. This method is considered a cost-effective and sustainable alternative to those currently used. This method requires low-energy inputs. However, it has its limitations, such as climatic conditions and difficulties in separating algae from the water. Thus, further research is needed on techniques for obtaining high microalgae biomass in order to apply the methods on a wider scale [256,257].

## 5. Plant Endophytes Resistant to TEs

Interactions between plants and soil microorganisms in phytoremediation have beneficial effects because it is an inexpensive method, and there is a low probability of harm to the environment [258,259]. Microorganisms inhabiting the internal tissues and intercellular spaces of plants without causing signs of pathogenesis are called endophytes [260]. Endophytes are commonly isolated from herbaceous plants [261,262,263]. The first study on the isolation of endophytic microorganisms resistant to TEs was published by Idris et al. [264]. The researchers, including Halácsy, isolated endophytic bacteria from the inside of a plant which is a hyperaccumulator of nickel—*Thlaspi goesingense*. The study was conducted in eastern Austria, where the total nickel content per kilogram of soil was 2.5 mg. The isolates were classified into two classes: *Alfaproteobacteria* and Gram-positive bacteria. About 42% of the isolates exhibited a high degree of similarity to the *Methylobacterium mesophilicum* species and 37% to *Sphingomonas* sp. Other isolates exhibited similarities to the following genera: *Rhodococcus*, *Curtobacterium*, and *Plantibacter*. El-Deeb et al. [265] isolated endophytic bacteria of the *Enterobacter* genus from an aquatic plant *Eichhornia crassipes*, common in Egypt. The bacterial strains exhibited resistance to zinc, cadmium, and lead. In 2008 Sun et al. [266] isolated endophytic bacteria from rapeseed (*Brassica napus*) growing in the suburbs of Nanjing, China. The soil from which the plants had been collected had the highest levels of lead (216.5 mg/kg) and zinc (204.5 mg/kg). Lead-resistant bacteria with predominant strains *Microbacterium* sp. and *Pseudomonas fluorescens* were extracted from the rapeseed. The researchers also found that the bacteria promoted plant growth because they produced plant hormones, dissolved lead, produced siderophores and 1-aminocyclopropane-1-carboxylic acid deaminase [267,268]. Ma et al. [269] found that *Sedum plumbizincicola* was not resistant to cadmium, zinc, or lead. The researchers isolated the following bacteria: *Achromobacter* sp., *Bacillus* sp., *Bacillus pumilus,* and *Stenotrophomonas* sp. In subsequent scientific studies, isolates of endophytic bacteria were found in the endosphere of plants growing in areas contaminated with TEs (Table 6).

Soil microorganisms can increase the solubility and oxidation of metals by releasing organic ligands, decomposing organic matter and secreting metabolites and siderophores [273,274]. Abou-Shanab et al. [275] observed that the presence of a specific microbiota increased the phytoextraction of nickel by the *Alyssum murale*. Low-molecular-weight organic acids produced by microorganisms, such as gluconic acid, 2-ketoglutarate, oxalate, citrate, acetate, malate and succinate, play a special role in the mobilization of TEs. Whiting et al. [276] found that the inoculation of soil with metal-resistant rhizosphere bacteria significantly increased the availability of zinc ions and their accumulation in plants. Siderophores—low-molecular-weight organic chelators with high affinity for iron ions Fe^3+^, synthesized by microorganisms in the presence of iron Fe^2+^ deficiency, play an important role in the mobility of metals. These compounds have relatively low selectivity and show an affinity for numerous metal ions—Al, Cd, Cu, Ga, In, Pb, and Zn [8,277,278]. The metals bound by bacterial siderophores can be taken up by bacteria and plants, thereby increasing the level of metal accumulation in plant tissues. A prime example is pyoverdine, synthesized by bacteria of the *Pseudomonas* genus.

## 6. Bacterial and Fungal Influence on Growth of Metallophytes

Microbial populations are known to influence the movement of TEs and their availability through the action of chelating agents, acidification, dissolution of phosphates and changes in redox conditions [222].

Bacteria can cause changes in the mobility of TEs, which facilitate their uptake by plants. The following bacterial species have this ability: *Bacillus* sp., *Escherichia coli, Pseudomonas putida, Thiobacillus ferrooxidans, Shewanella alga* and *Acinobacter* sp. [279,280]. Some bacteria exhibit a special tolerance to metallic elements because they have high adaptability to the environment or have special proteins allowing heavy metals to bind through chelates, thus reducing their toxicity [281]. Due to the fact that bacteria require different growth conditions, environmental factors often significantly influence heavy metal adsorption [282]. Wang et al. [283] found that when the pH value was too low, hydrogen ions competed with metal ions for adsorption sites on the bacterial surface. When the pH value was too high, metal ions and hydroxide ions formed hydrated hydroxide precipitation. The most favorable environmental pH for bacteria is 5–6, as the adsorption effect of heavy metal ions on the bacterial surface is the best. When the pH value is too low, heavy metal ions on the cell surface are desorbed from the cell, and thus, the adsorption capacity of the bacteria for heavy metals is limited [284].

Transformations occurring in the rhizosphere of metallophytes are fundamentally different from the processes observed in the root zone of plants. An example is the symbiosis of bacteria of the *Rhizobium* genus with legumes. Selenium hyperaccumulators such as *Astragalus bisulcatus, A. racemosus* and *A. praelongus* were observed to live in symbiosis with rhizobia, capable of growing in the presence of heavy metals [277,285]. The specific environment of the rhizosphere of metallophytes is a rich reservoir of metal-resistant microorganisms such as rhizosphere fungi and zinc hyperaccumulator bacteria, *Thlaspi caerulescens* [8]. Plant growth-promoting rhizobacteria (PGPR) are particularly noteworthy as this specific group of microorganisms can directly stimulate plant growth [212,286,287]. These mechanisms include the synthesis of compounds such as ACC deaminases, plant hormones such as auxins and the aforementioned secretion of siderophores and free nitrogen fixation. The promotion of metallophyte growth by rhizosphere bacteria increases the plant biomass where heavy metals are bound, thus increasing the efficiency of the phytoextraction process [212,288,289]. Phytoremediation is also facilitated by PGPR with bacterial auxin and indole-3-acetic acid (IAA), which stimulates lateral root growth and affects the development of root trichomes [60,287,290,291,292]. For example, rhizospheric IAA synthesized by both plants and bacteria can signal soil *Streptomyces* to increase antibiotic production. They inhibit bacterial and fungal phytopathogens and competing microbes [293]. Indole-3-acetic acid (IAA) is the main substance that significantly affects plant growth. IAA produced by rhizobia disrupts the aerial physiological processes altering the plant auxin pool [294]. Some bacteria stimulate plant growth by degrading plant-synthesized IAA when its levels are higher than normal [295]. IAA is an undoubted source of carbon and energy for bacterial growth and development. These microorganisms can use this plant attractant hormone and thus fight off competition [296]. The presence of metals in the soil usually interferes with the metabolism of other elements. Phosphorus in metal-rich soils occurs in bound and insoluble forms, such as polyphosphates and organic phosphorus compounds [8]. However, many metal-resistant PGPR can release soluble phosphorus, thus making it available to plants. Phosphate-solubilizing microorganisms produce gluconic acid, which is an intermediate product of the metabolism of various bacteria of the *Pseudomonas* and *Ervinia* genera [212]. Mycorrhiza can also play a significant role in the accumulation of heavy metals by plants. Arbuscular mycorrhizal fungi (AMF) are also worth mentioning, as they are another supporting source for plants involved in phytoremediation. Their presence increases the absorptive surface area of plant roots, and thus, the uptake of water and nutrients, as well as the availability of TEs, increases. AMF are able to produce phytohormones stimulating plant growth [210,297]. Javaid et al. [298] observed that AMF secreted glomalin—a protein forming complexes with metals. Mycorrhizal fungi receive growth substrates produced by photosynthesis from the plant. In return, they provide the plant with mineralized and available elements, such as P and Cu [8,299]. Mycorrhized plant roots gain additional absorbent surface area, which increases the efficiency of metal binding. *Berkheya coddi* is an excellent example of a mycorrhized plant. It can accumulate twenty times more Ni than non-mycorrhizal plants [300]. Rhizosphere microorganisms not only promote plant growth but also increase the pool of available metal ions in the soil. The activity of bioremediation processes conducted by soil microorganisms is the greatest and most effective in the rhizosphere. Research on the role of soil microbes involved in the remediation of soils contaminated with TEs may help to develop more effective technologies for removing heavy metals from this environment [301].

## 7. Conclusions

Soil pollution of TEs is a serious problem in the modern world. Unlike air or water pollution, soil-polluting TEs remain there much longer than other elements of the biosphere. All TEs in high concentrations are toxic to humans, animals, plants and microorganisms. Conventional soil remediation methods are often inapplicable, so it is necessary to intensively search for innovative and environmentally friendly techniques for ecosystem clean up using phytoremediation. Phytoremediation, referred to as green technology, is widely used to remediate soils contaminated with TEs and is used to treat sediments, groundwater and surface water. Like any method and this procedure has both advantages and disadvantages.

In recent years, great progress has been seen in improving the efficiency and quality of the phytoremediation process. This method, combined with burning the resulting biomass to produce heat and electricity, may prove to be one of the key techniques for environmental clean-up [302]. At this stage, it seems essential to create effective transgenic plants that are good phytoremediators. Thus, a huge challenge is to obtain genetically modified plants that will result in the ability to accumulate pollution in their large biomass. In the case of TEs, the preference is focused on aboveground parts that can then be easily harvested. Maintaining translocation from the root to the shoot, followed by sequestration in vacuoles and/or other parts of the cells of the plant’s aboveground organs, are the most commonly used strategies for genetic modification [9]. Genetically modified plants should also exhibit high viability and be more resistant to environmental stress, which will make them better competitors among native plant varieties. In addition, it is important for scientists to understand the mechanisms of natural phytoremediation, which is still not fully understood. Until these undiscovered mechanisms are clarified, the trial-and-error method seems to be the only reasonable tool [303]. Purification of soils on an industrial scale will most likely be possible in the future through the use of genetically modified organisms. It is estimated that over the next 25 years, the European Union will allocate about 100 trillion euros to clean up degraded areas [222]. It is, therefore, necessary to intensify the research being carried out in this direction in order to create a plant that can remove and accumulate these pollutants sparsely and in large quantities as soon as possible.

Today’s engineering bioremediation offers quite a few effective solutions in the form of the use of various organic substances (e.g., sewage sludge, sorbents, enzymatic and microbial preparations or nanoparticles). However, it is extremely important that the preparations or sorbents used do not adversely affect the environment and are easily and quickly biodegradable. This is because ignorance and unawareness of the far-reaching effects of their use can be a danger. The technique of assisting bioremediation with genetic engineering still arouses much controversy. There are a number of restrictions on its use. This is due to strict regulations and safety considerations. It should be remembered that there is always a significant risk of gene transfer from transgenic plants or microorganisms to the environment. Another huge drawback is that genetic research on microbiota and plants capable of efficient phytoremediation is usually conducted in specialized laboratories, which unfortunately does not reflect natural conditions. Great hope has been placed in international projects. One such project currently underway in Poland is the international ‘GOLD’ project called “Bridging the gap between phytoremediation solutions on growing energy crops on contaminated lands and clean biofuel production,” which has received sizable funding from Horizon in 2020. This project is very important because naturally polluted areas are being studied. This is because the above practices are translated into reality (two municipalities in Upper Silesia). Based on this research, universal strategies will be developed that can be applied to other potentially contaminated sites and used in various countries in the European Union and Asia. Thus, phytoremediation is becoming one of the elements of both an integrated and sustainable approach to the revitalization of polluted areas and the protection and shaping of the space in which we live.

The future of phytoremediation development must therefore involve the development of technologies for the utilitarian use of the biomass obtained. Remediation of polluted soil is time-consuming and, in hyperaccumulating plants, takes 2–60 years, while in non-hyperaccumulating plants, it takes 25–2800 years [230]. Phytoremediation may be a viable option for the removal of TEs contamination from environments, as the biomass created in the process could be economically used in the form of bioenergy [304].

A holistic approach is therefore needed to assess the effectiveness of phytoremediation, requiring the joint efforts of engineers, agronomists, plant biologists and microbiologists to work together with policy makers, regulators and industry representatives. Key tasks for phytoremediation are the valorization of phytoremediation biomass to offset remediation costs. In addition, it is clear that all stakeholders expect the creation of phytoremediators that will ensure that all risks are minimized while maximizing both economic, ecological and social benefits.

## Figures and Tables

**Table 1 plants-12-01653-t001:** Examples of hyperaccumulators and recommended phytoremediation methods.

Plant Species	TEs	Method	References
*Alyssum murale*	Ni	phytoextraction	[91,92,93]
*Alyssum pintodasilvae*	Ni	phytoextraction	[94,95]
*Arabidopsis halleri*	Cd, Zn	phytoextraction	[93,96,97]
*Azolla pinnata*	Cd, Zn, Ni	phytoextraction	[98,99]
	Cu	rhizofiltration	[100,101]
*Berkheya coddii*	Ni	phytoextraction	[92,93]
*Brassica juncea*	Pb, Cd, Cu, Ni, Zn, Cr	rhizofiltration	[93,102]
*Brassica oleracea*	Tl	phytoextraction	[95,103]
*Betula occidentalis*	Pb	rhizofiltration	[104]
*Cicer aeritinum* L.	Cr, Cu, Cd, Pb	phytoextraction	[105,106]
*Eichhornia crassipes*	Cu, Pb	rhizofiltration	[93,107]
*Eleocharis acicularis*	Cu, Cd, Zn, As, Pb	phytoextraction	[108,109]
*Euphorbia* sp.	Cu, As, Cd, Pb, Zn	phytostabilization	[93,110,111]
*Haumaniastrum robertii*	Co	phytoextraction	[101,112]
*Helianthus annuus*	Cd, Pb, Cr, Ni	phytostabilization	[32,113,114]
*Iberis intermedia*	Tl	phytoextraction	[95,103]
*Ipomoea alpina*	Cu, Hg	phytostabilization	[115,116]
*Jatropha curcas* L.	Cd, Cu, Ni, Pb, Hg, As	phytoextraction	[93,117]
*Lactuca sativa* L.	Cd, Pb	phytoextraction	[118,119,120]
*Lepidium sativum* L.	Cd, Pb, As	phytoextraction	[118,121]
*Macadamia neurophylla*	Mn	phytoextraction	[122]
*Miscanthus × giganteus*	Cu, Zn, Cd, Pb, Ni	phytoextraction	[123,124,125]
*Nicotiana tabacum*	Cd, Zn	phytoextraction	[95,126,127,128]
*Pisum sativum* L.	Cd, Cu, Cr, Co, Ni, Pb	phytoextraction	[129,130]
*Pelargonium* sp.	Pb	phytoextraction	[131,132]
*Pteris vittata*	As	phytoextraction	[95,133,134]
*Salix viminalis*	Cu, Zn, Pb, Cd	phytoextraction	[93,123,135,136]
*Salvia sclarea* L.	Cd, Zn, Pb	phytoextraction	[137,138]
*Spinacia oleracea* L.	Cd, Pb, As, Sb	phytoextraction	[118,139]
	Cd, Al	phytostabilization	[140]
*Thlaspi caerulescens*	Cd	phytoextraction	[93]
	Zn	rhizofiltration	[93,102]
*Thlaspi goesingense*	Ni	phytoextraction	[92,93]
*Tagetes minuta*	As, Pb	phytoextraction	[141,142]

**Table 2 plants-12-01653-t002:** Examples of hyperaccumulators and the TEs content in part of plant.

Plant Species	TEs	TEs Accumulation (mg kg^−1^)	TEs Accumulated Part of Plant	References
*Alyssum bertolonii*	Ni	10,900	Shoots	[145]
*Alyssum murale*	Ni	4730–20,100	Leaves	[92]
*Arabidopsis halleri*	Zn	5722	Shoots	[146]
*Azolla pinnata*	Cd	740	Roots	[98]
*Brassica juncea*	Zn	30,550	Roots	[147]
	Cd	25,000	Roots	[148]
*Eleocharis acicularis*	Cu	20,200	Shoots	[149]
	Zn	11,200		
	As	1470		
*Euphorbia cheiradenia*	Pb	1138	Shoots	[150]
*Pteris vittata*	As	8331	Frond and root	[151]
*Sedum alfredii*	Zn	9000	Leaves	[152]
*Thlaspi caerulescens*	Ni	6100	Rosette	[153]
	Zn	19,410	Leaves	[154]

**Table 3 plants-12-01653-t003:** Selected phytoremediation plant species of green roofs [159].

Plant Species	TEs	TEs Accumulation [mg kg^−1^]
*Ficus microcarpa*	Cd	419
	Cu	1260
	Pb	1050
	Zn	561
*Helichrysum italicum*	Zn	646 (root), 1176 (stem)
	Pb	346 (root), 484 (stem)
*Melastoma malabathricum*	Cd	426
	Cu	1820
	Pb	2390
	Zn	1380
*Pennisetum purpureum*	Cd	1.30–7.05 (stem)
*Portulaca grandiflora*	Pb	9.77
*Portulaca oleracea*	Cr (VI)	4600 (root), 1400 (stem)
*Sedum alfredii*	Cd	4512 (stem), 3317 (leaf)
*Sedum plumbizincicola*	Cd	35 (root), 93 (stem)
	Zn	889 (root), 1072 (stem)
	Pb	99 (root), 101 (stem)
*Solanum nigrum*	Cd	35.9 (root), 77.0 (stem), 117.2 (leaf)
	Zn	167.9 (root), 95.4 (stem), 85.5 (leaf)
	Cu	64.0 (root), 12.3 (stem), 32.2 (leaf)

**Table 4 plants-12-01653-t004:** Some chelators used in phytoextraction.

Plant Species	TEs	Chelator	References
*Arabidopsis* *halleri*	As, Hg	Thiol-rich chelators	[71]
*Brassica juncea*	Cd, Cu, Ni, Pb, Zn	Gallic and citric acid	[184]
	Cd, Cu, Pb, Zn	EDTA	[185]
	Cd	Citric acid and NTA	[186]
	Cr, Ni	EDTA, DTPA Oxalic acid, citric acid	[187]
	Au, Ag	NH_4_SCN	[188]
*Helianthus annuus*	Cu, Zn	EDDS	[189,190]
*Lolium perenne*	Cr, Ni, Zn	EDTA	[191]
*Phalaris arundincacea*	Cr	EDTA	[191]
*Thlaspi* *caerulascens*	Cd, Cr, Ni	EDTA	[192]
*Thlaspi* *goesingense*	Pb	[S,S]-ethylene diamine disuccinate	[193,194]
*Zea mays*	Zn	NTA	[195]

EDDS—ethylenediaminedisuccinic acid; EDTA—ethylenediaminetetraacetic acid; NTA—nitrilotriacetic acid; DTPA—diethylene triamine pentaacetic acid; NH_4_SCN—ammonium thiocyanate.

**Table 5 plants-12-01653-t005:** List of energy crops used in phytoremediation with consideration of bioenergy [229].

Bioenergy Crop	Soil Pollutants	Sustainable Bioenergy Production
*Jatropha curcas*	*Heavy metals*	Biodiesel (seed oil)
*Populus* spp.	*Organics, heavy metals*	Bioethanol (biomass)
*Salix* spp.	*Organics, heavy metals*	Bioethanol (biomass)
*Arundo donax*	*Organics, heavy metals*	Bioenergy, bioethanol (biomass)
*Miscanhtus*	*Organics, heavy metals*	Bioethanol (biomass)
*Ricinus communis*	*Organics, heavy metals*	Biodiesel (biomass and seed oil)
*Zea mays*	*Heavy metals*	Bioenergy (biomass)
*Halianthus annuus*	*Heavy metals*	Bioenergy, bioethanol (biomass and seed oil)
*Brassica* spp.	*Heavy metals*	Biofuel, biodiesel (seed oil)
*Canabis sativa*	*Heavy metals*	Bioenergy (biomass)

**Table 6 plants-12-01653-t006:** Resistance of endophytic bacteria growing in heavy metal contaminated areas.

Type/Species Endophytic Bacterium	Source of Bacteria Isolation	Resistance of Bacteria to TEs	References
*Achromobacter* sp.	*Sedum plumbizincicola*	Zn, Cd, Pb	[269]
*Acinetobacter* sp.	*Elsholtzia splendens*	Cu	[266]
*Bacillus* sp.	*Alnus firma Sedum plumbizincicola*	Zn, Cd, Pb	[269]
*Enterobacter* sp.	*Eichhornia crassipes*	Zn, Cd, Pb	[265,270]
*Methylobacterium mesophilicum*	*Thlaspi goesingense* Halácsy	Ni	[264]
*Microbacterium* sp.	*Brassica napus*	Zn, Cd, Cu, Ni, Pb	[271]
*Plantibacter* sp.	*Thlaspi goesingense* Halácsy	Ni	[264]
*Pseudomonas* sp.	*Alyssum serpyllifolium*	Ni	[269]
*Rhodococcus* sp.	*Thlaspi goesingense* Halácsy	Ni	[264]
*Serratia marcescens*	*Pteris vittata*	V	[272]

## Data Availability

Data sharing is not applicable to this article as no new data were created or analyzed in this study.

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
