# Peer review of "Phytoremediation as an Effective Remedy for Removing Trace Elements from Ecosystems"

_plants, 2023, doi:10.3390/plants12081653_

Round 1

Reviewer 1 Report (Previous Reviewer 3)

The authors properly revised the manuscript based on the comments from the reviewer. Some minor issues need to be further demonstrated.

1.       The “Ecosystems” in the title might be replaced by “Soils”.

2.       Line 95: The title is too simple and not appropriate.

Author Response

Reviewer 2 Report (New Reviewer)

Regarding the manuscript entitled "Phytoremediation as an Effective Remedy for Removing Trace Elements from Ecosystems" I have some comments which can be found in the attached. My main concern is describing the novelty of study, on the other hand many reviews have been published so far in this topic, please in abstract and introduction describe the novelty, what is the difference (please see https://doi.org/10.1016/C2020-0-02583-1, https://doi.org/10.3389/fpls.2020.00359, https://doi.org/10.1007/s42452-021-04301-4,  https://doi.org/10.3390/plants12030429, https://doi.org/10.3390/plants11091255, ....)

please in introduction describe the databases used, with the last access data search, and other criteria of searching strategy.

Round 2

Reviewer 2 Report (New Reviewer)

Although authors have revised the manuscript according to the suggested comments/recommendations, my main concern is still remained: what is the difference between the present study and previously published papers? What is new information? Why those are not cited and referred? 

Round 3

Reviewer 2 Report (New Reviewer)

Authors have revised items requested, in my opinion it can be considered for further publication procedure 

This manuscript is a resubmission of an earlier submission. The following is a list of the peer review reports and author responses from that submission.

Round 1

Reviewer 1 Report

The submitted manuscript is suitable for Plants, but in my opinion, it doesn't have quality in the current version as a review. It's too general, the mix terms such as metallophytes and hyperaccumulators. Table 1 could be more interesting with maximum levels and also how metal is accumulated. More in roots, shoots, etc. Also, I'm sure that they are more plant species that are reported on Table 1. For example, tobacco or sunflower are real hyperaccumulator plants that even can be used with radionuclides, and they're missing from this Table. It's not clear which criteria were followed for each question.

I have other comments that should also be reviewed.

L13-14, 20-21, L34, 48-70. I suggest avoiding HM and using Potentially Toxic Elements, because, in this review, Arsenic is also covered.

L37-38. I agree with this sentence, but PAHs are also a recalcitrant contaminant and not too easy to be degraded. I suggest rewriting this question.

L40-45. I suggest to rewrite this section. "The most important sources in soil include bedrock, industrial and traffic emissions, municipal services, and agriculture" and "The anthropogenic sources of heavy metal contamination of soils include mining and metallurgy of non-ferrous metals, the metallurgical industry, the chemical industry, landfilling, high-dose use of..." Is the same information. I suggest writing. "Their sources can be divided into natural origin (mainly parent material) and anthropogenic origin (mining, metallurgy, etc.)...

Besides, if the authors will write each source, i suggest writing also military and shooting activities since these activities are the second source of Pb in several countries. See https://doi.org/10.1155/2013/158764 (and plants can be used for phytorremediation https://link.springer.com/chapter/10.1007/978-3-319-40148-5_17 https://doi.org/10.1080/10934520903467832 and sometimes the roots can help to transform Pb to other Pb-forms https://doi.org/10.1007/s11356-015-5340-7)

L77. "They are called metallophytes or hyperaccumulators" This is not true. Metallophytes are organisms that can tolerate metal contamination, but hyperaccumulators are organisms that can tolerate and they have very good growth in contaminated areas. A plant can be metallophyte but not hyperaccumulator, since to be hyperaccumulator, plant part/soil content be > 50-100. 

L79. Table 1. This is not a systematic review. A very low number of papers. Probably it should be better that maximum metal levels should be highlighted and if metals are accumulated on roots or shoots. 

L84. And usually, they have a commercial interest.

L91. "cost-effective" I'm not sure about this term because phytoremediation measures require several years or decades.

L93-94. Only two? Sure?

In general, all sections need several improvements. Information is too general and very scarce. A table with the best species and the potential elements for each methodology will probably be more informative and useful for readers than in the present version. 

Sometimes it seems a rewriting of a book or other review paper. Several papers and reviews have been published with the manuscript topic, but the authors also need to give us any novelty. It could also be interesting to indicate the potential effects on biometric or oxidative stress parameters. Ok, we have a plant that it's very good for phytoremediation, but they can have adverse issues such as metal accumulation on grain (for example, wheat or rice), or also grow well but with lower biomass production. 

Reviewer 2 Report

The work is very well prepared and deserves to be published quickly.

I have no comments on the substantive side of the work.

Reviewer 3 Report

The manuscript attempted to review the basic research methods of phytoremediation, their strengths and weaknesses, the effects of microorganisms on metallophytes, and plant endophytes resistant to heavy metals. The topic is of practical significance for the management of trace elments contaminated environment. However, the manuscript was not well organised and prepared. Not much new information was provided and no systematic summaries and definite conclusions were given, resulting a superficial review. Future research prospects were not demonstrated at the end. In addition, more recent relevant studies published in classical journals should have been reviewed and cited.